# Cell-Level Spatio-Temporal Model for a Bacillus Calmette–Guérin-Based Immunotherapy Treatment Protocol of Superficial Bladder Cancer

**DOI:** 10.3390/cells11152372

**Published:** 2022-08-02

**Authors:** Teddy Lazebnik

**Affiliations:** Department of Cancer Biology, Cancer Institute, University College London, London WC1E 6DD, UK; t.lazebnik@ucl.ac.uk

**Keywords:** agent-based simulation, spatial biological model, cancer treatment, genetic algorithm, computer simulation, personalized clinical treatment

## Abstract

Bladder cancer is one of the most widespread types of cancer. Multiple treatments for non-invasive, superficial bladder cancer have been proposed over the last several decades with a weekly Bacillus Calmette–Guérin immunotherapy-based therapy protocol, which is considered the gold standard today. Nonetheless, due to the complexity of the interactions between the immune system, healthy cells, and cancer cells in the bladder’s microenvironment, clinical outcomes vary significantly among patients. Mathematical models are shown to be effective in predicting the treatment outcome based on the patient’s clinical condition at the beginning of the treatment. Even so, these models still have large errors for long-term treatments and patients that they do not fit. In this work, we utilize modern mathematical tools and propose a novel cell-level spatio-temporal mathematical model that takes into consideration the cell–cell and cell–environment interactions occurring in a realistic bladder’s geometric configuration in order to reduce these errors. We implement the model using the agent-based simulation approach, showing the impacts of different cancer tumor sizes and locations at the beginning of the treatment on the clinical outcomes for today’s gold-standard treatment protocol. In addition, we propose a genetic-algorithm-based approach to finding a successful and time-optimal treatment protocol for a given patient’s initial condition. Our results show that the current standard treatment protocol can be modified to produce cancer-free equilibrium for deeper cancer cells in the urothelium if the cancer cells’ spatial distribution is known, resulting in a greater success rate.

## 1. Introduction and Related Work

Cancer is a generic name for a wide range of illnesses in which cells in a specific part of the body grow and reproduce uncontrollably. This uncontrolled reproduction can make the cancerous cells invade and destroy surrounding healthy tissue, leading to other diseases [1]. One form of cancer is bladder cancer (BC), which is considered to be a very aggressive form of cancer by the World Health Organization, with 16,641 and 5457 deaths in 2018 in the United States and the United Kingdom, respectively (a full data report can be found at https://platform.who.int/mortality/themes/theme-details/topics/indicator-groups/indicator-group-details/MDB/bladder-cancer (accessed on 4 July 2022)). In the same year, 549,393 individuals worldwide suffered from BC, 125,311 (22.80%) of whom were female and 424,082 (77.19%) of whom were male [2].

Multiple treatments of non-invasive (superficial) BC have been proposed, including but not limited to chemotherapy [3] and immunotherapy [4]. From these options, the immunotherapy treatment was found to be the most promising one, following a larger shift in oncological treatment protocols [5,6]. Nonetheless, immunotherapy treatments for BC have not advanced much over the last several decades after the treatment protocol suggested by [7], which involves weekly instillations of Bacillus Calmette–Guérin (BCG). The BCG treatment is based on an attenuated non-pathogenic strain of Mycobacterium bovis that was originally used as a vaccine against tuberculosis [8,9,10]. Despite its promising clinical outcome, a non-negligible portion of the patients receiving standard BCG treatments are affected in unwanted ways, including lack of clinical response or relapse after several months [11]. One way to tackle this challenge is to optimize the treatment protocol in general and for the patient’s initial conditions in particular [12]. This requires a more accurate representation of the patient’s state before the BCG treatment and a more accurate evaluation of the course of treatment.

In order to predict the course of treatment, one can use mathematical models and computer simulations, which have been shown to be powerful tools in clinical settings in general and oncology in particular, allowing for the investigation of both diseases and possible treatment protocols [13,14,15]. Indeed, researchers developed multiple models to describe BCG-based immunotherapy treatment of BC [16,17,18]. Specifically, one dominant group of models is that of ordinary differential equation (ODE)-based models that describe cell population dynamics [19,20]. These models describe the changes in several cell populations over time due to temporal interactions between these cell populations. Commonly, these models include immune system cells, tumor cells, and treatment-affected cells [21]. Moreover, several attempts were made to improve these models by introducing spatial dynamics that would take into account the biological interactions in a physical space based on partial differential equations [22,23]. In particular, Ref. [24] extended the model proposed by [25] by introducing a ring–sphere geometrical configuration with diffusion dynamics to all population dynamics based on the spatial model proposed by [26]. In a similar manner, Ref. [27] further extended the model proposed by [24] by introducing more detailed biological dynamics based on the biological description proposed by [19], increasing the number of cell populations from four to 10. The authors showed that the extended model provides a more realistic treatment protocol prediction compared to that of the same model without spatial dynamics. Moreover, Ref. [28] provided a comparison between the ring–sphere and the more realistic ring–ellipsoid geometric configuration, revealing even more accurate results for the latter. These results were not a surprise, as in vivo experiments revealed the prognostic significance of tumor location for survival outcomes in patients with BC [29,30]. Hence, to obtain the optimal treatment protocol using a mathematical model, one is required to take into consideration the geometrical configuration of the bladder and the location of the cancer polyps during the course of the treatment.

While these models are of great promise to healthcare professionals, they neglect the spatial information of cell–cell interactions in the bladder microenvironment. As such, by taking these processes into consideration, one can obtain a more realistic and accurate representation of the biological dynamics occurring for the BCG-based immunotherapy treatment of BC, further improving the efficiency of the treatment and saving lives. In this work, we propose an agent-based simulation approach that extends the mathematical model proposed by [31] by introducing spatial components to the cells that take into consideration their relative locations in the bladder, as well as movement dynamics. Hence, the novelty of this work is two-fold. First, we introduce healthy cells and their interactions with cancer, the BCG-based treatment, and the immune system, thus allowing a spatio-temporal simulation of interactions at the cell–cell level. Second, we use a GA-based search approach to find the optimal BCG immunotherapy treatment based on the patient’s spatio-temporal properties, obtaining an optimal solution on all of the treatment parameters at once. It is of note that, as far as we know, previous models for BCG treatment for BC have not been clinically validated; therefore, a benchmark of their performance is not available. As such, we could not empirically compare the performance of the proposed model relative to the previous models.

The remainder of the paper is organized as follows: In Section 2, we introduce our spatio-temporal mathematical model of BCG-based treatment of BC. In Section 3, we present a numerical simulation of the proposed model for several clinical stages and a numerical sensitivity analysis for several parameters by using an agent-based simulation approach. In Section 4, we offer a method for finding the time-optimal treatment protocol given the patient’s initial condition by using a genetic algorithm. In Section 5 and Section 6, we discuss the main clinical results arising from the model and propose final remarks with possible future work. A schematic view of this structure is presented in Figure 1.

## 2. Model Definition

The human bladder consists of luminal or superficial, basal, and intermediate cell zones, each of which contains a variable number of cell layers and helps us to deduce the definition of every stage of disease by following the regions in which non-papillary or papillary tumors grow [32]. In particular, superficial BC includes low-grade papillary transitional cell carcinoma (stages Ta and T1) and carcinoma in situ [31]. Growth occurs superficially on the inner surface of the bladder in the form of a polyp, but does not extend to the muscle [33]. For superficial BC, local surgery is applied first, followed by intravesical BCG treatment. In this treatment, bacterial installations are introduced into the bladder with a catheter (inserted through the urethra). Afterward, BCG accumulates next to the bladder wall and in superficial cancer cells. While binding to the cell wall, BCG is incorporated by both antigen-presenting cells and uninfected cancer cells, which ultimately kill the BCG-infected cancer cells and eliminate the entire tumor [34,35].

Our modeling approach has similar temporal dynamics to those of [36,37,38,39,40] in the sense that we use a cell population approach to describe the biological interactions among the different cell populations and between them and the environment. In particular, we extended the ODE description proposed by [31], which includes three cell populations: BCG-infected cancer cells, uninfected cancer cells, and effector (immune system) cells, as well as an equation to describe the concentration of BCG in the bladder over time, operating as a geometry-free environment. However, in order to model the bladder’s tissue precisely, we introduce healthy cells and BCG-infected cell populations. Hence, effector cells (*E*) destroy BCG-infected cancer cells ($C_{i}$) at an average rate of $p_{3}$. In parallel, BCG-infected cancer cells become infected with BCG (*B*) at an average rate of $p_{2}$ [41]. In addition, the population of uninfected cancer cells exponentially increases at an average rate of *r* [25]. BCG is captured by effector cells (*E*), which reduce the BCG’s concentration at an average rate of $p_{1}$ [42], in addition to a natural decay with an average half-life of $\mu_{B}^{-1}$. The effector cells are recruited from the bone marrow by infected cancer and healthy cells at an average rate of α. At the same time, encounters between effector cells and BCG result in further recruitment of effector cells at an average rate of $p_{4}$, while encounters between them and BCG-infected cancer (and non-cancer) cells result in elimination of effector cells at an average rate of $p_{5}$. Similarly to BCG, effector cells naturally decay with an average half-life of $\mu_{E}^{-1}$. Healthy cells ($H_{u}$) are looking to fill the bladder’s geometry, which requires $H^{m}$ healthy cells. It is assumed that healthy cells have a maximum reproduction rate of $p_{6}$. Healthy cells, like uninfected cancer cells, are infected by BCG at an average rate o f$p_{7}$ and become BCG-infected cells $H_{i}$. BCG-infected cancer cells recruit effector cells at an average rate of β and are destroyed by them at an average rate o f$p_{8}$. Finally, the BCG is injected periodically, following the treatment protocol proposed by [43]. From the spatial point of view, we adopt the mesh obtained by [44] for the outer layer of the bladder. Following the assumption presented by [27], we copied the same mesh for the inner layer such that the average Euclidean distance ($L_{2}$) between the outer and inner layers is constant and is equal to ψ. In addition, based on the model proposed by [45], we used eight cell layers of the urothelium with a depth of 40 cells. A three-dimensional view of the geometrical configuration of the urothelium with a schematic representation of the layers is shown in Figure 2.

The geometrical configuration of the model is populated by agents according to the following procedure: (1) Uninfected cancer cells ($C_{u}$) are introduced to a random location (*L*; or a given location defined by the user) next to the inner layer of the bladder, forming a perfect sphere as a function of the number of cells and their average size; (2) effector cells (*E*) are uniformly distributed in the bladder. (3) The remaining geometry of the bladder is filled with healthy cells (*H*). The BCG-infected cancer cell population size is always zero at the beginning of the simulation, as no BCG has been injected yet. At the beginning of the simulation, a location (Γ) is defined to be the point of injection of BCG into the bladder such that Γ is located on the inner layer of the bladder in the place in which the pre-surgery took place. Furthermore, an injection of the BCG as part of the treatment is affected by three additional parameters: the amount of BCG injected each time (*b*), the injection rate (τ), and the number of injections (*k*). Formally, one can represent the dynamics as a system of partial differential equations, as shown in the following.

In Equation (Equation 1), $\frac{\partial E(t,\bar{x})}{\partial t}$ is the dynamical rate of the effector cell population’s distribution over time and space. It is affected by the following six terms. First, effector cells are eliminated by BCG at a rate of $p_{4}$. Second and third, BCG-infected cancer and healthy cells are killed by effector cells at a rate of $p_{5}$. Fourth, effector cells are recruited by the immune system at a rate of α by the BCG-infected cancer cell population. Sixth, effector cells are recruited by the immune system at a rate of β by the BCG-infected healthy cell population.
(1)$$\begin{array}{c} \frac{\partial E(t,\bar{x})}{\partial t}=\left(p_{4}B\left(t,\bar{x}\right)-p_{5}C_{i}\left(t,\bar{x}\right)-p_{5}H_{i}\left(t,\bar{x}\right)-\mu_{E}\right)E\left(t,\bar{x}\right)+\alpha C_{i}\left(t,\bar{x}\right)+\beta H_{i}\left(t,\bar{x}\right). \end{array}$$

In Equation (Equation 2), $\frac{\partial C_{u}\left(t,\bar{x}\right)}{\partial t}$ is the dynamical rate of the uninfected cancer cell population’s distribution over time and space. It is affected by the following two terms. First, uninfected cancer cells grow in a diffusive manner with an exponential rate of *r*. Second, uninfected cancer cells become BCG-infected cancer cells after an interaction with BCG at an average rate of $p_{2}$.
(2)$$\begin{array}{c} \frac{\partial C_{u}\left(t,\bar{x}\right)}{\partial t}=\left(r\frac{\partial^{2}C_{u}\left(t,\bar{x}\right)}{\partial {\bar{x}}^{2}}-p_{2}B\left(t,\bar{x}\right)\right)C_{u}\left(t,\bar{x}\right). \end{array}$$

In Equation (Equation 3), $\frac{\partial C_{i}\left(t,\bar{x}\right)}{\partial t}$ is the dynamical rate of the uninfected cancer cell population’s distribution over time and space. It is affected by the following two terms. First, uninfected cancer cells become BCG-infected cancer cells after an interaction with BCG at an average rate of $p_{2}$. Second, effector cells eliminate BCG-infected cancer cells at an average rate of $p_{3}$.
(3)$$\frac{\partial C_{i}\left(t,\bar{x}\right)}{\partial t}=p_{2}\frac{\partial^{2}C_{i}\left(t,\bar{x}\right)}{\partial {\bar{x}}^{2}}B\left(t,\bar{x}\right)C_{i}\left(t,\bar{x}\right)-p_{3}E\left(t,\bar{x}\right)C_{u}\left(t,\bar{x}\right).$$

In Equation (Equation 4), $\frac{\partial H_{u}\left(t,\bar{x}\right)}{\partial t}$ is the dynamical rate of the uninfected healthy cell population’s distribution over time and space. It is affected by the following two terms. First, uninfected healthy cells aim to occupy the entire bladder’s geometry and reproduce at an average rate of $p_{6}$. Second, uninfected healthy cells become BCG-infected healthy cells after an interaction with BCG at an average rate of $p_{7}$.
(4)$$\begin{array}{c} \frac{\partial H_{u}\left(t,\bar{x}\right)}{\partial t}=p_{6}H_{u}\left(t,\bar{x}\right)\left(1-\frac{H_{u}\left(t,\bar{x}\right)+H_{i}\left(t,\bar{x}\right)+C_{i}\left(t,\bar{x}\right)+C_{u}\left(t,\bar{x}\right)}{H_{m}}\right)-p_{7}B\left(t,\bar{x}\right)H_{u}\left(t,\bar{x}\right). \end{array}$$

In Equation (Equation 5), $\frac{\partial H_{i}\left(t,\bar{x}\right)}{\partial t}$ is the dynamical rate of the BCG-infected healthy cell population’s distribution over time and space. It is affected by the following two terms. First, uninfected healthy cells become BCG-infected healthy cells after an interaction with BCG at an average rate of $p_{7}$. Second, effector cells eliminate BCG-infected cancer cells at an average rate of $p_{8}$.
(5)$$\begin{array}{c} \frac{\partial H_{i}\left(t,\bar{x}\right)}{\partial t}=p_{7}B\left(t,\bar{x}\right)H_{u}\left(t,\bar{x}\right)-p_{8}E\left(t,\bar{x}\right)H_{u}\left(t,\bar{x}\right). \end{array}$$

In Equation (Equation 6), $\frac{\partial B(t,\bar{x})}{\partial t}$ is the dynamical rate of the BCG cell population’s distribution over time and space. It is affected by the following six terms: first, the diffusion rate of the BCG cell population inside the bladder’s geometry; second, a quantity *b* of BCG instilled into the bladder every τ days. As the installation of the BCG is modeled by a shifted Dirac delta function δ(t−nτ),n∈{0,1,…,N−1}, the $n_{th}$ dose raises $B(t,\bar{x})$ by *b* units at t=nτ. The third term is the elimination of BCG by effector cells according to an average rate $p_{1}$; fourth, the BCG tumor cell grows with a rate coefficient $p_{2}$. Fifth, the BCG is eliminated by BCG-infected healthy cells according to the rate coefficient $p_{7}$. Sixth, the bacteria cells die with a rate coefficient of $\mu_{\beta }$.
(6)$$\begin{array}{c} \frac{\partial B(t,\bar{x})}{\partial t}=d\frac{\partial^{2}B\left(t,\bar{x}\right)}{\partial {\bar{x}}^{2}}+\sum_{n=0}^{k}b\delta \left(t-n\tau ,\Gamma \right)-\left(p_{1}E\left(t,\bar{x}\right)+p_{2}C_{u}\left(t,\bar{x}\right)-p_{7}H_{u}\left(t,\bar{x}\right)-\mu_{B}\right)B\left(t,\bar{x}\right). \end{array}$$

In addition, the following initial condition is defined:(7)$$\begin{array}{c} E\left(0,\bar{x}\right)=e,\;C_{i}\left(0,\bar{x}\right)=0,\;C_{u}\left(0,S\left(L,\sqrt[{3}]{\frac{3\xi }{4\pi }}\right)\right)=\xi >0,\;B\left(0,S\left(\Gamma ,1\right)\right)=b>0,\;H\left(0,\bar{x}\right)=H_{m}-\xi , \end{array}$$
where S(l,z) is a sphere located in *l* with a radius of *z*; ξ is the initial number of uninfected cancer cells. A schematic view of the cell–cell interactions and cell-environment interactions in the proposed model is presented in Figure 3. Notably, the explicit formalization of the parameters’ functions is challenging to either obtain or define; in practice, it would be solved by a spatially local numerical approximation, leaving only temporally fixed-term parameters.

## 3. Numerical Simulation

### 3.1. Method

The proposed model is solved using the agent-based simulation approach [46,47] such that each cell is treated as an agent that is described by a finite-state machine [48]. Namely, each cell, *c*, is described by its type ($\tau \in \{E,C_{u},C_{i},H_{u},H_{i}\}$) and location in the bladder’s geometry $\bar{x}$. For simplicity, we assume that the cells have a box shape with a fixed volume. No two cells can occupy the same space. As such, each cell is surrounded by other cells or by the geometry’s border. For each step in time, during the cell reproduction step, generated cells appear next to the cell that generated them and cause a change in the bladder’s geometry by elastically pushing all of the cells around them. In addition, we assume that effector cells do not have volume and can travel inside the tissue freely. As such, for the cell level, Equations (Equation 1)–(Equation 6) can be approximated using a discrete ODE and simulated as follows. First, each uninfected healthy and uninfected cancer cell has a probability $p_{2}$ or $p_{7}$, respectively, of becoming infected by BCG, multiplied by the number of BCG cells next to them (and up to some radius ζ). Only after the cell dies can the BCG occupy its spatial position and travel further into the urothelium. BCG-infected cancer and regular cells locally interact with the effector cells next to them and up to some radius ζ. In addition, BCG-infected cancer and regular cells are also naturally excluded from the urothelium. Furthermore, as healthy cells are generated, they are generated from the first (most shallow) layer and push the layers above it to fill the free spaces of previously excluded cells. The simulation stops with the condition $min_{t\in N}\min T,C_{i}\left(t\right)=C_{u}\left(t\right)=B\left(t\right)=H_{i}\left(t\right)=0$. Table 1 summarizes the default parameter values used in the simulation with their respective sources. These values can be treated as an approximation over the entire bladder’s geometrical configuration.

A tumor-free equilibrium is obtained when $C_{i}\left(t,\bar{x}\right)=C_{u}\left(t,\bar{x}\right)=0$, indicating that there are no more cancer cells in the bladder. This is a necessary condition, but not enough to obtain a successful treatment, which also requires that not too many BCG cells are located inside the bladder at any point in time. However, the second condition can be ignored if the total number of injected BCG cells is below the threshold.

### 3.2. Results

We evaluated if, when the classical treatment protocol defined by Γ was set to be as close to the tumor as possible (*k* = 6, *b* = 1.07 $\times 10^{6}$, and τ=10,080 min (e.g., one week)) [43], it could result in a tumor-free equilibrium for various initial conditions involving different cancer cell population sizes and spatial distributions. For that, we computed *n* = 100 simulations for the configuration of the initial uninfected cancer population size ($C_{u}\left(0,\bar{x}\right)$) and the location of the center of mass of this cell population as a function of the urethral layer. The results are shown in Figure 4, where the value is the portion of simulations that ended with a tumor-free equilibrium, the X-axis is the layer of the urothelium in which the uninfected cell population’s center of mass is located, and the Y-axis is the size of this population. Moreover, we used a least mean square [50] with symbolic regression [51] to find the function that best fit the obtained data while being as simple as possible (i.e., a function is considered simpler if it has fewer terms and a lower polynomial degree), resulting in:(8)$$f\left(s,C_{u}\right):=1.459-0.118C_{u}-0.158s-0.177sC_{u},$$
where *s* is the layer of the cancer cells’ center of mass in the urothelium. Equation (Equation 8) was obtained with a coefficient of determination of $R^{2}=0.94$.

In addition, the sensitivity of the model to changes in the treatment parameters with respect to the default treatment protocol was explored, as shown in Figure 5, where the X-axis provides the parameter values and the Y-axis represents the normalized treatment performance. Namely, the normalized treatment performance is the number of successful treatments obtained by applying the modified treatment protocol divided by the number of successful treatments obtained with the classical treatment protocol for a given initial condition. Specifically, the polar distance from the cancer cell population’s center of mass monotonically decreases, as shown in Figure 5a. On the other hand, the number of BCG injections, number of injected BCG cells, and duration between any two consecutive BCG injections increase up to some point (7, 1.2, and 6) and then decrease, as presented in Figure 5b–d, respectively.

## 4. Time-Optimal Treatment Protocol

The BCG immunotherapy treatment is defined by four parameters that can be controlled by the clinician: the BCG injection location (Γ), the number of injections (*k*), the duration between every two injections (τ), and the number of BCG cells injected each time (*b*). While the number of BCG cells injected each time can also differ in order to further improve the treatment protocol, it practically requires conducting additional tests on the patient [52]; therefore, it is not within the scope of the current model. In addition, there are two uncontrolled parameters: the location of the cancerous tumor and its size (i.e., $C_{u}\left(0,\bar{x}\right)$).

Clinicians commonly look to find a time-optimal and successful treatment protocol in which the patient is healthy after the course of the treatment in a minimal amount of time [27]. In this work, we adopt the definition of a successful treatment protocol proposed by [27]: $\exists t^{*}\in N:C_{u}\left(t^{*},\bar{x}\right)=0\wedge \forall t\in N:B\left(t,\bar{x}\right)\leq 10^{8}$, and the time-optimal treatment protocol is the treatment protocol that satisfies $\min_{t^{*}}:C_{u}\left(t^{*},\bar{x}\right)=0$. In order to find a successful and time-optimal protocol, one can define the task as a search problem that explores the space of all possible treatment protocols and looks for the treatment protocol that satisfies both conditions.

### 4.1. Method

One approach to accomplishing this task is the use of a genetic algorithm (GA). GAs are a family of search (and optimization) methods based on the biological theory of evolution [53]. GAs have been widely used in different forms for numerical simulations [54,55] and global search tasks [56,57,58,59], providing promising results. Formally, a GA simulates the process of “evolving” through natural selection, where genes (in our case, treatment protocols) that obtain a better score (i.e., are more adapted) from the fitness function have a higher probability of passing their genes on to the next generation of genes. In between every two generations, the GA performs three stochastic processes: mutation [60], crossover [61], and a feasibility test [62]. In particular, for our case, a gene *g* is defined by a tuple g:=(Γ,k,τ,b) corresponding to the treatment’s parameters. As the initial condition, the GA gets a random population of genes of size A>>1. For each generation, each gene passes a mutation operator that picks an element of the gene at random (uniformly distributed) and alters its value by introducing a new sample from a normal distribution with a mean value of 0 and a standard deviation of $\left(SD_{\Gamma },SD_{k},SD_{\tau },SD_{b}\right)$[53]. Afterward, a crossover operator gets two genes from the population at random and returns two new genes using the *ring* crossover method [63]. Namely, given genes $g_{1}$ and $g_{2}$ and a random index 0<i<3, the new genes $g_{1}^{*}$ and $g_{2}^{*}$ are defined by $g_{1}^{*}=g_{1}^{0,\cdots i}\cup g_{2}^{i+1,\cdots 3}$ and $g_{2}^{*}=g_{2}^{0,\cdots i}\cup g_{2}^{i+1,\cdots 3}$. Next, the genes’ fitness is computed by solving the numerical simulation of the proposed model and returning $\min_{t^{*}}:C_{u}\left(t^{*},\bar{x}\right)=0$ if such $t^{*}$ exists and −∞ elsewhere. Finally, we use the “tournaments with royalty” selection operator [61]. Formally, the gene with the best fitness score in each generation is kept for the next generation, and the gene that has a better fitness probability will lead to further better genes in the following generations. Hence, some portion of the population, θ∈(0,1), is kept for the next generation, taking the genes with the highest fitness scores. Afterward, the remaining portion of the next generation’s population is populated by selecting genes with a probability corresponding to their normalized fitness score (normalized such that the sum of all remaining genes’ fitness scores is 1). The GA stops either after a pre-defined number of generations (υ∈N) or if the average fitness score of the genes’ population between two successive generations is lower than a pre-defined threshold Δ<<1.

Intuitively, a GA is used as a global search method. It starts at a random population and moves over generations to a better population with a better fitness function, in a stochastic manner. Thus, GAs are usually classified as directed-search algorithms [64]. This process does not promise a global optimum [65]. Nonetheless, one can remedy this limitation by running the GA multiple times with different initial conditions and taking the best outcome [66], as was done in this study.

### 4.2. Results

Using the proposed GA, we computed the time-optimal treatment protocol for multiple patients’ initial conditions with different cancer cell population distributions and sizes. Since the treatment was assumed to be provided on a daily basis, we constrained the duration between two BCG injections to be a fixed number of days. The results of this analysis are shown in Figure 6, where the X-axis is the layer of the urothelium in which the uninfected cell population’s center of mass is located and the Y-axis is the size of this population; the value represents the predicted duration of the time-optimal treatment protocol. A black cell in the heatmap indicates that the algorithm was not able to find a successful treatment protocol after 100,000 iterations in a test of *n* = 1000 random samples.

## 5. Discussion

In this work, we present a novel spatio-temporal mathematical model of BCG immunotherapy in non-invasive, superficial BC that takes into consideration cell–cell and cell–environment interactions; this was implemented by using an agent-based simulation approach on top of a realistic three-dimensional geometrical configuration of the urothelium. The proposed model focuses more on a computational improvement over the original model [31] rather than including more biological parameters, as other studies have attempted [24,27]. Hence, the novelty of the proposed model lies in two computational improvements: the usage of a mesh-based, three-dimensional, spatio-temporal, agent-based simulation of the dynamics and the usage of a GA-based search approach in order to find the optimal treatment protocol by optimizing all of the treatment’s parameters given a personalized spatio-temporal initial condition of the patient. In addition, as far as we know, this is the first model to take healthy cells, the spatial influence of cancer cells, and the spatial spread of BCG cells into consideration.

Based on this model, we evaluated the effectiveness of the classical treatment protocol [43], which is considered the gold standard to date [25], to obtain a tumor-free equilibrium for different spatial configurations of the size and distribution of cancer cells, as shown in Figure 5. Unsurprisingly, with greater shallowness and fewer cancer cells, the the treatment protocol worked better. In particular, based on Equation (Equation 8), the initial number of cancer cells is responsible for 26% of the treatment success rate, while the average layer in the urothelium is responsible for 35%, and the combination between the two is responsible for 39%, as one can find by normalizing the coefficients of the parameters (not including the free coefficient). Hence, according to the proposed model, the classical treatment protocol, which does not have any specifications for the cancer cells’ spatial distribution, is effective for either a large mass of cancer cells located on a very shallow layer of the urothelium or a much deeper layer when the number of cancer cells is relatively small. These results agree with those of previous models [24,25]. Nonetheless, due to the lack of clinical evaluations of these or previous models, it is challenging to conclude on the exact accuracy of these predictions.

Moreover, when computing the sensitivity of the controlled parameters of the treatment protocol, one can see that the classical treatment protocol is close to optimal on average, as shown in Figure 5. Nonetheless, different configurations, such as seven BCG injections rather than six or 1.20 $\times 10^{6}$ injected BCG cells compared to 1.07 $\times 10^{6}$ (Figure 5b,c), show slightly better results. A possible explanation for this phenomenon is the fact that the treatment protocol’s performance is computed for a single geometrical configuration of a woman [44] (the original treatment is a design based on a majority of male participants). Thus, Figure 5 reveals the advantage of a personalized treatment, even for a single parameter of the treatment protocol at a time. In addition, Figure 5 provides a sanity check for the cell–cell interactions; as BCG cells are injected further and further from the urothelium’s inner tissue in general and from cancer cells in particular, they find fewer ways to spread in the urothelium, which is in agreement with the outcomes proposed by [7].

Based on the conclusion that the treatment protocol should be adapted based on the initial spatial conditions of each patient, we proposed a GA approach in order to optimize the treatment protocol. This allows one to find a successful and time-optimal treatment protocol (see Section 4). When using the GA proposed for our case, on average, one can obtain the shortest duration of a successful treatment protocol, as shown in Figure 6. In particular, one can notice that the GA was able to introduce cases in which the classical treatment protocol was found to fail, as shown in the comparison of Figure 4 and Figure 6. In addition, to obtain a similar treatment success rate for the cases in which the cancer is located in a deeper layer of the urothelium, the treatment should be longer. However, after a certain point, no feasible treatment protocol exists. This outcome stands in opposition to the results reported by [28], who claimed that a treatment protocol existed for even the eighth layer. The difference in the predictions is rooted in the lack of healthy cells that are generated during the treatment and that occupy the urothelium’s geometry. This results in more aggressive BCG injections during the treatment in order to make sure that the BCG cells will reach these deeper layers, thus overshooting the BCG threshold of a successful treatment protocol. Hence, the proposed model better agrees with the experimental results obtained by [67] regarding the limitations of the BCG-based immunotherapy treatment for BC compared to previous models [26,28,31].

Since BCG immunotherapy is currently the most effective therapy for superficial BC, but a non-negligible portion of patients do not react to the treatment [68], we believe that healthcare professionals can use our model to provide a better, more personalized treatment based on relatively simple-to-obtain spatial data on the cancer cell population’s size and distribution. For instance, one application for the proposed model is to aid in computing the appropriate injection rate of BCG cells in a given clinical situation. As part of a scan of the area, three-dimensional imagery can be used to compute an approximated initial condition for the proposed model. Using these data, the model would provide a suggestion for a personalized, optimal treatment protocol for a given patient. In a similar manner, if the model predicts that the BCG immunotherapy treatment will not succeed, clinicians can explore other treatment protocols to obtain a better outcome.

A limitation of the proposed work is that the results were obtained for a single geometrical configuration [44], which may cause a bias. In addition, the proposed results are not clinically validated, as with the previous mathematical models, and they are based on previous clinical, biological, and mathematical theories based on related empirical research conducted on subjects. As a result, the model’s de facto performance is unknown and should be evaluated using in vivo data from clinical experiments before one can broadly use it in clinical practice. Similarly, a comparison of the proposed model with other models designed to improve the treatment protocol is infeasible at the moment due to the lack of benchmark data. Until this issue is resolved, in future research, these mathematical models can be extended to take more biological processes and more accurate spatial and temporal dynamics into consideration in order to make the models more expressive.

## 6. Conclusions

Mathematical modeling of clinical treatment protocols for bladder cancer is considered a powerful tool for better understanding the mechanisms of tumor growth and response to therapy [21,22]. Thus, it opens the door for the development of better (and more personalized) treatment protocols. Hence, models that more accurately represent biological and clinical dynamics are able to provide a more accurate prediction of a treatment protocol’s outcome. In this study, we showed that, by taking realistic spatial properties of the bladder and the cancer cell population into consideration, one is able to obtain a finer description of the bioclinical dynamics. These preliminary results can be exploited afterward by healthcare professionals and can be used to fine-tune the treatment protocol given the required data. For instance, during the BC diagnosis phase, pet-CT (computerized tomography) is commonly used. The image obtained from the pet-CT can provide the information needed to use the proposed model. This means that no additional tests are required to utilize the proposed model. Therefore, possible future work would involve the integration of a computer-vision-based solution that receives a pet-CT scan and converts it into the spatial configuration used by the proposed model, which was inspired by the method proposed by [69]. In a similar manner, the evaluation of the proposed model’s performance in clinical settings should be tested with an in vivo experiment that compares the classical treatment protocol and the one described by the proposed model. 

## Figures and Tables

**Figure 1 cells-11-02372-f001:**
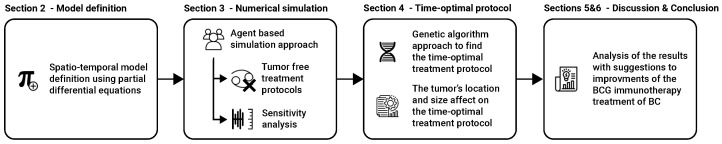
The manuscript’s structure. First, we introduce our novel cell-level spatio-temporal model. Afterward, an agent-based simulation approach is proposed to numerically solve the model, and the tumor-free treatment protocol based on the current standard treatment protocols is evaluated, in addition to the sensitivity of the treatment’s performance to changes in its parameters. Next, a genetic-algorithm-based approach to finding a time-optimal treatment protocol is described and used to explore the feasibility of a BCG-based treatment of different initial conditions. Finally, an analysis of the results and closing remarks are provided.

**Figure 2 cells-11-02372-f002:**
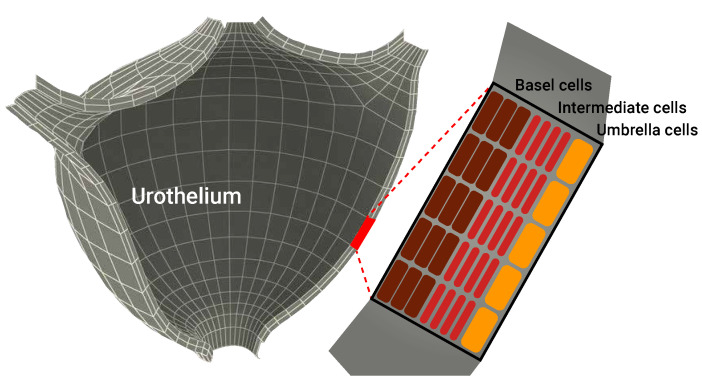
A schematic view of the geometrical configuration of the urothelium used in the model. The three-dimensional mesh configuration was adopted from [44]. In addition, the urothelium’s geometry is populated by eight layers of cells with a depth of 40 cells, following the model proposed by [45].

**Figure 3 cells-11-02372-f003:**
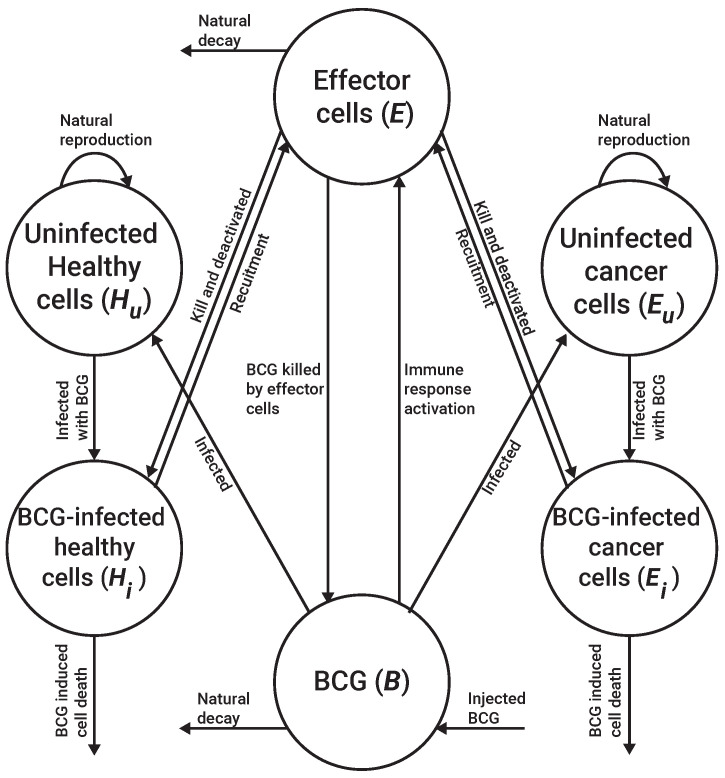
A schematic view of the cell–cell interactions and cell–environment (BCG) interactions in the proposed model.

**Figure 4 cells-11-02372-f004:**
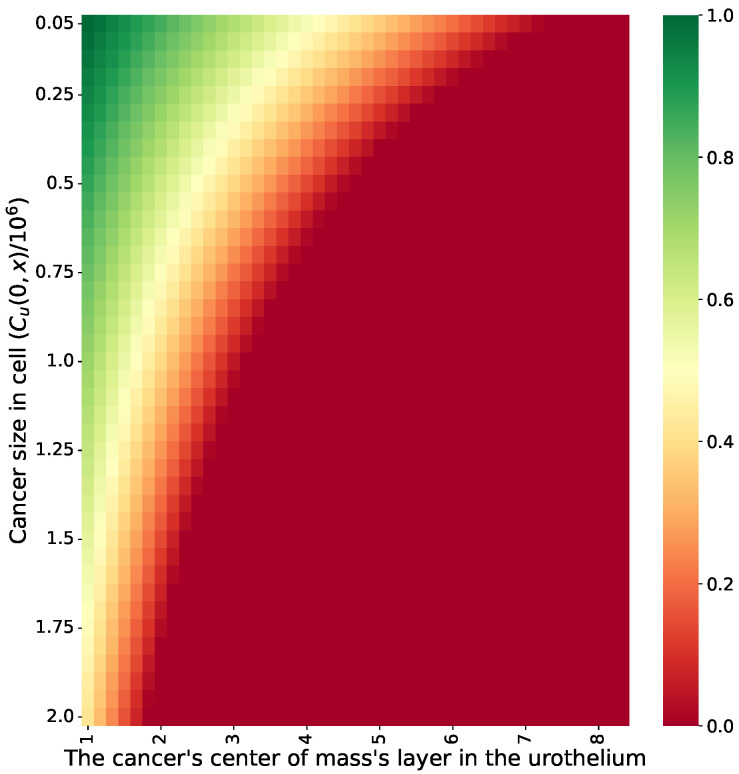
The portion of treatments that resulted in a tumor-free equilibrium for *n* = 1000 reparations as a function of the initial uninfected cancer cell population size and the layer in which the cancer cell population’s center of mass was located. The results are shown for the standard treatment protocol shown in Table 1.

**Figure 5 cells-11-02372-f005:**
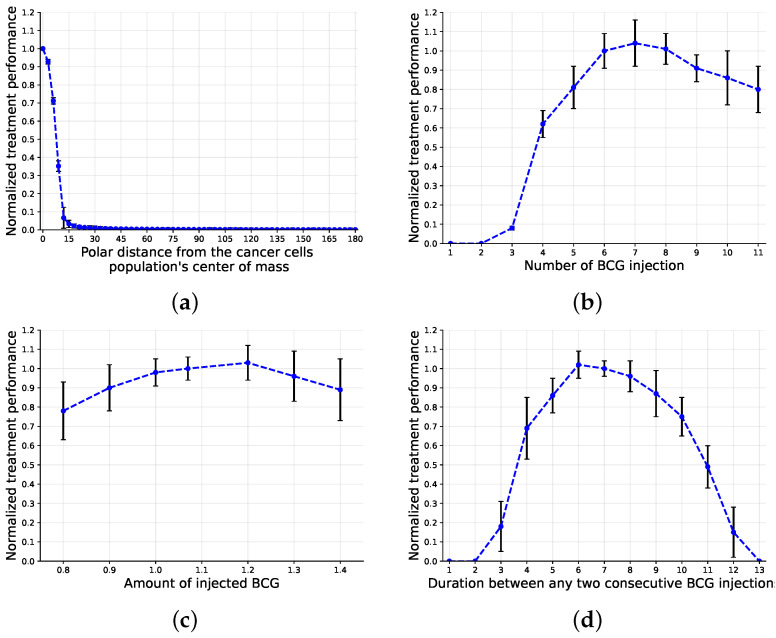
Sensitivity analysis of the four controlled parameters of the treatment protocol. The classical treatment protocol is based on the one proposed by [43] and provided in Table 1. The results are normalized to the classical treatment protocol and are shown as the mean ± standard deviation for *n* = 1000 repetitions. (**a**) BCG injection location (Γ); (**b**) number of BCG injections (*k*); (**c**) number of BCG cells injected (*b*); (**d**) duration between any two consecutive BCG injections (τ).

**Figure 6 cells-11-02372-f006:**
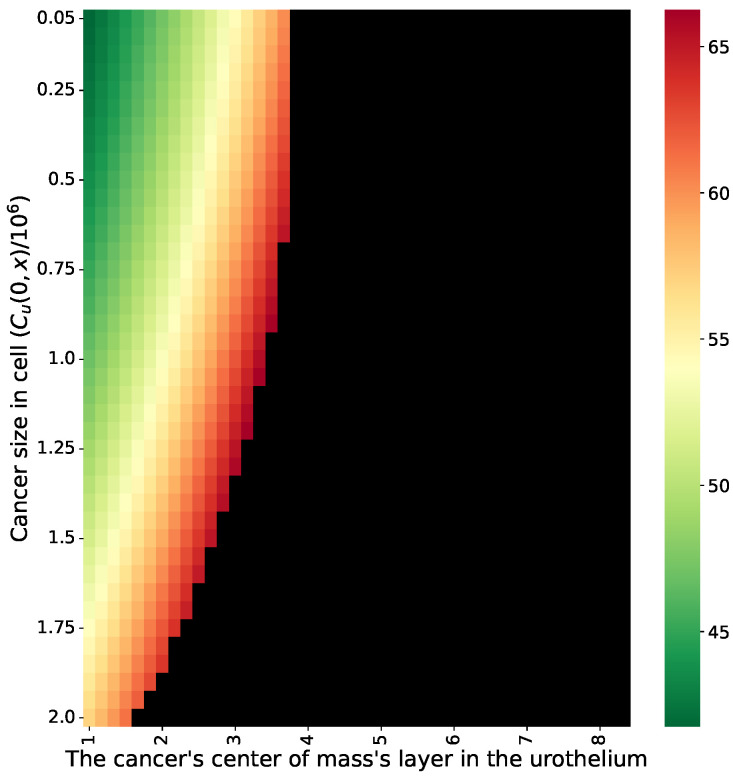
The average duration in days of successful and time-optimal treatment for n=1000 reparations as a function of the initial uninfected cancer cell population size and the layer in which the cancer cell population’s center of mass is located. Black cells indicate that the GA failed to find a successful treatment protocol. The results are shown for the parameter values provided in Table 1.

**Table 1 cells-11-02372-t001:** The default values of the model’s parameters’ values and sources.

Parameter	Symbol	Value	Source
The rate of BCG cells killed by effector cells [1]	$p_{1}$	1.25 $\times 10^{-7}$	[42]
Infection rate of tumor cells by BCG [1]	$p_{2}$	2.85 $\times 10^{-6}$	[31]
Rate of destruction of BCG-infected tumor cells by effector cells [1]	$p_{3}$	1.09 $\times 10^{-7}$	[36]
The immune system’s activation response rate [1]	$p_{4}$	1.20 $\times 10^{-6}$	[31]
The rate of effector cell deactivation after binding with BCG-infected tumor cells in days [1]	$p_{5}$	3.45 $\times 10^{-10}$	[36]
Production rate of uninfected healthy cells in days [1/t]	$p_{6}$	0.37	[49]
Rate of destruction of BCG-infected healthy cells by effector cells [1]	$p_{7}$	1.09 $\times 10^{-7}$	[36]
Destruction rate of BCG-infected cancer cells by effector cells [1]	$p_{8}$	1.1 $\times 10^{-8}$	[32]
Average effector cell decay rate in days [1/*t*]	$\mu_{E}$	4.1 × $10^{-1}$	[31]
Average BCG decay rate in days [1/*t*]	$\mu_{B}$	0.1	[31]
The number of BCG cells injected [1]	*b*	1.07 $\times 10^{6}$	[43]
The duration, in days, between any two consecutive BCG injections [*t*]	τ	7	[43]
Rate of effector cell stimulation due to infected tumor cells in days [1/*t*]	α	5.2 $\times 10^{-3}$	[25]
Average carrying capacity of an uninfected tumor cell [1]	β	1.1 $\times 10^{-8}$	[25]
Tumor growth rate in days [1/*t*]	*r*	1.22 $\times 10^{-2}$	[49]
Upper boundary of the number of healthy of cells in the urothelium [1]	$H_{M}$	1.4 $\times 10^{12}$	[45]
Number of BCG injections in the standard treatment protocol [1]	*k*	6	[43]
Polar Euclidean ($L_{2}$) distance from the location of uninfected cancer cells’ center of mass in meters [m]	Γ	0	[43]
The average Euclidean distance between the outer and inner mesh of the bladder [1]	ϕ	40 cells	[45]
The number of effector cells in the urothelium at the beginning of the treatment [1]	*e*	$10^{5}$	Assumed
Stop condition threshold for the best gene’s fitness score in two consecutive generations of the genetic algorithm [1]	Δ	0.05	Assumed
The genetic algorithm’s mutation rate [1, 1, 1, 1]	$SD_{\Gamma }$, $SD_{k}$, $SD_{\tau }$, $SD_{b}$	0, 0.05, 0.05, 0.05	Assumed
Number of generations for the genetic algorithm [1]	υ	100	Assumed
Population size of the genetic algorithm [1]	$GA_{s}$	100	Assumed [1]
Royalty rate for the genetic algorithm [1]	θ	0.05	Assumed [1]
Number of times that the GA algorithm runs with different initial conditions for each input [1]	*z*	15	Assumed

## Data Availability

All of the data used in this research are available online and are cited in the text.

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
