# Peer review of "Cell-Level Spatio-Temporal Model for a Bacillus Calmette–Guérin-Based Immunotherapy Treatment Protocol of Superficial Bladder Cancer"

_cells, 2022, doi:10.3390/cells11152372_

Round 1

Reviewer 1 Report

Title: Cell-level Spatio-Temporal Model for Bacillus Calmette–Guérin Based Immunotherapy Treatment Protocol of Superficial Bladder Cancer

Respected Editor

After carefully reading the manuscript, I found that paper can be accepted, if the authors carefully handle these comments

1) Highlight the novelty of the paper

2) It is better to write the references as [a-b] or [a,b], choose one of style.

3) Use proper punctuations in the whole paper

4) Use commas or full stop at the end of each equation

5) Genetic algorithm is used as a global search method in this study?

6) Is genetic algorithm is used as optimization procedures?

7) what are the stopping criteria of the genetic algorithm?

8) How did you choose the classical treatment protocol values?

9) In Eq. 3,  what is the meaning of  .s, which is written at the end?

10) For the numerical results, if possible provide at least one table in this study

11) Use the recent references of genetic algorithm, I suggest to choose the recent work of Prof. Muhammad Umar that he used for the biological models and genetic algorithm. You can find this (https://scholar.google.com/citations?hl=en&user=rjDQ8b0AAAAJ&view_op=list_works&sortby=pubdate) 

Author Response

First of all, we would like to thank the reviews for the careful review and for making sure we produce the best manuscript we can. It is honestly very appreciated to get such a detailed review. For your convenience, all text introduced due to the comments is highlighted in bold. 

Reviewer #1

Comment 1: “Highlight the novelty of the paper”

Answer 1: Thank you for this suggestion. Following this comment, we highlighted the novelty of this work in the Introduction section, stating: 

Hence, the novelty of this work is two-folded. First, we introduce healthy cells and their interactions with cancer, BCG-based treatment, and immunity system; allowing a spatio-temporal cell-cell level interaction simulation. Second, we use a GA-based optimization approach to find the optimal BCG immunotherapy treatment based on the patient's spatio-temporal properties, obtaining an optimal solution on all the treatment parameters at once.

Comment 2: “It is better to write the references as [a-b] or [a,b], choose one of style.”

Answer 2: Thank you for this proposal. You are right and following this comment, we make sure all the references in the manuscript are of the format [a,b,...,z]. Moreover, we used the journal’s Latex template to make sure we are using the right style format requested by the journal.

Comment 3: “Use proper punctuations in the whole paper”

Answer 3: Thank you for pointing our attention to this shortcoming. Following this comment, we asked a native English speaker to proofread the paper and make sure we use punctuations properly in addition to minor grammatical improvements. Thus, we hope now the paper is free from grammatical errors. Nonetheless, if accepted, we will work closely with the journal’s editing team to make sure the camera-ready version will be of high quality. 

Comment 4: “Use commas or full stop at the end of each equation”

Answer 4: Thank you for the suggestion. Following this comment, we fixed this issue.

Comment 5: “Genetic algorithm is used as a global search method in this study?”

Answer 5: Thank you for this question. First of all, the answer is yes. A genetic algorithm is used as a global search method. The authors are familiar with the fact that a genetic algorithm is not promised to provide a global optimum but the existence of such a global optimum is not in the scope of this study. Moreover, genetic algorithms used in the past for global search, obtaining decent results. Following this comment, we extended the introduction to the genetic algorithm in section 4 and introduce text that convey several points: a) the algorithm is used as a global search method. b) how we used the monte-carlo wrapper to make sure the result is robust. c) cite several relevant studies to show this approach is feasible (see also comment 11). We hope this modification well address the reviewer’s comment properly.

Comment 6: “Is genetic algorithm is used as optimization procedures?”

Answer 6: Thank you for this question. A genetic algorithm is by definition a directed search algorithm. However, if it searches for a configuration that minimizes a loss function, it is also can be treated as an optimization procedure. In this manner, the answer is yes in the context of this study. However, we do not treat it as an optimization procedure but as a search one, in order to be aligned with how health professionals would find the solution as the paper is addressed to a mixed audience of clinical/bioinformatics and mathematical/computer sciences. Following this comment, and the previous one, we highlighted this manner in section 4.1.

Comment 7: “what are the stopping criteria of the genetic algorithm?”

Answer 7: Thank you for this question. The classical stopping criteria and used in this manuscript is the number of generations provided to the genetic algorithm. Following this question, we better explain it in section 4.1  

Comment 8: “How did you choose the classical treatment protocol values?”

Answer 8: Thank you for this question. The classical treatment protocol values are taken from a clinically tested, currently considered the general golden standard. As such, the values are taken from previous publications and cited in Table 1. This table was in the appendix but following this and other comments we moved it into text as it seems it is important to understand the results.  

Comment 9: “In Eq. 3,  what is the meaning of  .s, which is written at the end?”

Answer 9: Thank you for pointing out this issue. The original meaning was to indicate this is a multiplication between the two numbers. However, we see the confusion and altered the equation’s text to be “0.177sC_u” to better convey the formula. 

Comment 10: “For the numerical results, if possible provide at least one table in this study”

Answer 10: Thank you for this suggestion, since the tables we were able to think about are huge and therefore cannot be included in a written form, we would add them as complementary materials to the manuscript and address them in the Discussion section. 

Comment 11: “Use the recent references of genetic algorithm, I suggest to choose the recent work of Prof. Muhammad Umar that he used for the biological models and genetic algorithm. You can find this (https://scholar.google.com/citations?hl=en&user=rjDQ8b0AAAAJ&view_op=list_works&sortby=pubdate) ”

Answer 11: Thank you very much for this suggestion - it was indeed very helpful. Following this comment, we introduce several references to section 4.1 to better convey to the reader the usefulness of the genetic algorithm in clinical settings.  

Reviewer 2 Report

The abstract includes introduction and method. It is expected to mostly contain significant, preferably quantitative, results.

In the method section, the parameters are described in the text altogether through lines 104-133; to make it more readable it is better to describe the parameters of each equation just before or next to the equation.

Unfortunately, the equations are presented in the manuscript very carelessly and I seriously doubt their correctness and whether they are implemented correctly and accurately: 

Why the parameters are assumed to be r and t dependent in the equations, while they are assumed to be constant?

I don't understand the first terms in equations for Cu an Ci. Why is the diffusion term mixed with the growth term? Where is the diffusion coefficient? 

Probably a Hu term s missing from the first term of the PDS equation for Hu cells.

Where are the units in table A1?

What do you mean by 1 in the term B(0, S(Gamma,1)) in equation 2? Are the parameters dimensionless?

Fig 3 is better be moved before the equations.

What are ksi and small s in equation 2? They should be defined prior to the equation.

What is f in equation 3?

Results and method sections are not well separated in the "numerical simulation" and "Time-optimal Treatment Protocol" sections.

"Normalized treatment performance" is not well defined in the text.

The GA algorithm is also not well described. What are SDs values for the normal distributions in GA?

Did the authors provided the code of the simulations and the optimization algorithm in the supplementary?

About the discussion: It should be well clarified what is improved in the current work over previous studies. Is there any new prediction in agreement with clinical results? Any novel proposed protocol? What is improved by the agent-based model instead of a continuous PDE model? Can the authors compare these methods and discuss the advantages.  Furthermore, the relationship between the ABM and the PDE equations is not clear in the method section.

Author Response

First of all, we would like to thank the reviews for the careful review and for making sure we produce the best manuscript we can. It is honestly very appreciated to get such a detailed review. For your convenience, all text introduced due to the comments is highlighted in bold. 

Reviewer #2

Comment 1: “ The abstract includes introduction and method. It is expected to mostly contain significant, preferably quantitative, results.”

Answer 1: Thank you for pointing out this shortcoming. Following this comment, we extended the abstract to include more explanation of the results and the significance of this work.  

Comment 2: “In the method section, the parameters are described in the text altogether through lines 104-133; to make it more readable it is better to describe the parameters of each equation just before or next to the equation.”

Answer 2: Thank you for this suggestion - you are absolutely right. Since Cells’ audience is more biological and clinical oriented compared to a more mathematical or computer science audience, we tried to avoid a long technical description. Nonetheless, addressing this comment, a full description of the equation, including a paragraph with a detailed, technical description before each equation is now provided after the short description, as now shown in the end of Section 2.

Comment 3: “Unfortunately, the equations are presented in the manuscript very carelessly and I seriously doubt their correctness and whether they are implemented correctly and accurately: Why the parameters are assumed to be r and t dependent in the equations, while they are assumed to be constant?”

Answer 3: Thank you very much for this remark. The parameters “r”, “\tau_E”, and  “tau_B” are time and spatial depending on the cell level however, as such multiplied by the cell population one cannot remove the (t, x) notation from them as you suggested. Thus, following this comment we removed the (t, x) notation and better convey this idea in the the new description of the equations provides in section 2. 

Comment 4: “I don't understand the first terms in equations for Cu and Ci. Why is the diffusion term mixed with the growth term? Where is the diffusion coefficient? ”

Answer 4: Thank you for this question. The growth term and the diffusion are connected to each other. Let us explain: on the cell level, the cell population is growing as each cell in the population is divided. As the division of cells is a spatial-local phenomenon, it is limited by the environment of each cell independently. Thus, we combined the growth rate with the diffusion parameter to compute the ability of cells to divide into more cells. As the growth rate is usually the average of the population, it is can be treated as a constant and thus the coefficients of the diffusion. On the other hand, the diffusion dynamics are dependent on the environment and thus promise that new cancer cells would be introduced to the system around previous cancer cells. Following this and other comments, we better explain the ABS implementation of the PDE representation, which sheds more light on the numerical simulation, as shown in section 3.1.

Comment 5: “Probably a Hu term s missing from the first term of the PDS equation for Hu cells.”

Answer 5: We are really sorry, but we were not able to understand the question. If the reviewer can explain the question, we would love to properly address it.

Comment 6: “Where are the units in table A1?”

Answer 6: Thank you for pointing our attention to this issue. Following this comment, we added the units to the table in the “Parameter” column.

Comment 7: “What do you mean by 1 in the term B(0, S(Gamma,1)) in equation 2? Are the parameters dimensionless?”

Answer 7: Thank you for this question. The term “1” stands for the radius of the BCG injected in cells. Since we assume in this research BCG is without volume, we needed to specify how it is located in the system at the beginning of the dynamics. 

Comment 8: “Fig 3 is better be moved before the equations.”

Answer 8: Thank you for this comment - fixed. Moreover, if the paper would be accepted, we will work with the journal’s editing team to make sure the figures are located properly in the final version.

Comment 9: “What are ksi and small s in equation 2? They should be defined prior to the equation.”

Answer 9: Thank you very much for addressing this issue. Following this comment, we clearly state that \xi is the initial number of an uninfected cancer cells. We removed “small s” as this was the average volume of the cancer cell in the initial phases of the implementation. However, in the implementation, we used to produce the provided results we assume the volume of cells is identical as described in section 3. Thus, we would sincerely thank the reviewer for spotting this error and alerting us about it.

Comment 10: “What is f in equation 3?”

Answer 10: “f” in Eq. (3) is the fitting function. Following this comment, we added the “:” before the equal sign (i.e., “:=”) to indicate we defining “f” in Eq. (3). One can also remove “f(s, C_u) := “ entirely but we find it easier to read and better convey the idea one can set the values in the fitting function (f) and obtain an approximation to the dynamics. Thus, we address the reviewer’s question with the first option. 

Comment 11: “Results and method sections are not well separated in the "numerical simulation" and "Time-optimal Treatment Protocol" sections.”

Answer 11: Thank you for this comment. Following this comment, we introduced sub-section headers to better divide the sections and altered the text accordingly to better define the line between the method used and the results obtained using the method. 

Comment 12: “Normalized treatment performance" is not well defined in the text.”

Answer 12: Thank you for this remark. Following this remark, we introduced into section 3.2 (see sub-section introduction following comment #11) the following statement in order to better explain the meaning of “Normalized treatment performance”:

…and the y-axis is the normalized treatment performance. Namely, the normalized treatment performance is the number of successful treatments obtained by applying the modified treatment protocol divided by the number of successful treatments obtained the classical treatment protocol.

Comment 13: “The GA algorithm is also not well described. What are SDs values for the normal distributions in GA?”

Answer 13: Thank you for this comment. First of all, following this comment, we introduced the values of the SDs parameters into Table 1. In addition, to address this comment, we introduce more explanation of how the genetic algorithm is working in our context in section 4.1 (see sub-section adding following comment #11).

Comment 14: “Did the authors provide the code of the simulations and the optimization algorithm in the supplementary?”

Answer 14: Thank you for this query. Indeed, upon acceptance, we will share the code with technical documentation in a GitHub repository. Moreover, we will share the data generated during this research so other researchers would be able to continue to investigate from the latest point without the need to use the code if they do not want to. 

Comment 15: “About the discussion: It should be well clarified what is improved in the current work over previous studies. Is there any new prediction in agreement with clinical results? Any novel proposed protocol? What is improved by the agent-based model instead of a continuous PDE model? Can the authors compare these methods and discuss the advantages.  Furthermore, the relationship between the ABM and the PDE equations is not clear in the method section.”

Answer 15: Thank you for these suggestions. Following this comment, we modified the Discussion section in a pronounced form. In particular, we alter the text to include:

  1. A clear novelty statement explaining the contribution of this model in comparison to previous models. 
  2. The fact that one can obtain a personalized treatment protocol using an initial condition in the form of the cancer cells spatial spread at the beginning of the treatment.
  3. The advantage of using ABS over PDE representation - in addition, we better explain in Section 3.1 how the ABS is implemented to approximate the PDE. 

Moreover, while we would love to compare our results with previous models, the lack of a clinical benchmark does not allow us to do so. We highlighted this limitation of the study and suggested it as future work. 

Round 2

Reviewer 1 Report

The authors have carefully revised all of my suggestions. The paper can be accepted in its present form

Author Response

Reviewer #1

Comment 1: “The authors have carefully revised all of my suggestions. The paper can be accepted in its present form”

Answer 1: We would like to thank once again the reviewer for his valuable comments that helped us to significantly improve the quality of our paper.

Reviewer 2 Report

Thanks to the authors for addressing the comments. I also learned some interesting points from them. Some remaining comments:

I think the first term in Hu equation should be multiplied by Hu.

Authors can simply remove the equations in eqn 1 and the parameter description above it as they are well discussed in the following.

I didn't find the definition of x_bar. Is it dimensionless?

I still think expressing the time and space dependency of the parameters is misleading as they are constant in time and space, as stated in the parameters table.

Units are not still well introduced in the table. t is not unit it is the time dimension. Is it second, minute, day?

Can you emphasize any new result, for the novel method, that better fits or explain the already existing clinical or experimental findings. It could better describe the advantages of the relatively complex new method. Personally I belive that when there is not better description for the previous experimental data, there isn't a well defined problem to address by modeling. Whether the model is complex or simple.

Author Response

First of all, we would like to sincerely thank once again the reviewer for his/her careful review, which significantly helped us to improve our work. 

Reviewer #2

Comment 1: “ I think the first term in Hu equation should be multiplied by Hu.”

Answer 1: We would like to thank the reviewer for this suggestion. Firstly we tried to explain why this suggestion is not working on the cell level which we try to capture but while carefully contracting the counter-example we notice the reviewer is absolutely right. Thus, we sincerely thank the reviewer for noticing this shortcoming and alerting us about it, allowing us to make sure the proposed model is accurately put. Following this comment, we introduce the needed modification to equation (4). 

Comment 2: “Authors can simply remove the equations in eqn 1 and the parameter description above it as they are well discussed in the following.”

Answer 2: Thank you for this suggestion. Following this comment, we removed Eq. (1) and moved the descriptive definitions of the equation to in its place. We left the introduction section for a more biological- and clinical-oriented audience, such as most of “cells” readers while we agree it is explained in detail in the equation description. We hope the reviewer would agree with our suggested editorial view for this section.

Comment 3: “I still think expressing the time and space dependency of the parameters is misleading as they are constant in time and space, as stated in the parameters table.”

Answer 3: Thank you very much for this suggestion. After careful thought, we agree with the reviewer and removed the (t, x) annotation. While the original idea was to show that these parameters can be (and biologically, are) time and space dependent, these hide inside biological processes that do not take into consideration in the current model, and thus it is redundant to show this annotation. Moreover, the reviewer is right that in practice we used a constant approximation to these values due to a lack of more detailed biological knowledge. Thus, we altered equations 1 to 6, removing the (t, x) annotation from the parameters. 

Comment 4: “Units are not still well introduced in the table. t is not unit it is the time dimension. Is it second, minute, day? ”

Answer 4: Thank you for this comment. All the time-depended parameters are reported in days and the distance parameter (\Gamma\) is reported in meters. Following this comment, we introduce the text “in days” / “in meters” in each of the relevant rows in Table 1. 

Comment 5: “Can you emphasize any new result, for the novel method, that better fits or explain the already existing clinical or experimental findings. It could better describe the advantages of the relatively complex new method. Personally I belive that when there is not better description for the previous experimental data, there isn't a well defined problem to address by modeling. Whether the model is complex or simple.”

Answer 5: Thank you for this question. We agree with the reviewer. While we would like to propose a comparison to existing clinical or experimental findings this is a challenge as they use a single method that produces the same results in the last three decades. In order to evaluate the proposed model, one is required to perform an in vivo experiment and to use our model to decide on a treatment protocol. Nonetheless, following this comment we searched for the closest argument, we can confidently report. Since one of the two main outcomes of our model is that the proposed model with the GA algorithm is able to state which treatment won’t work for different spatial configurations of the cancer cells, we searched for any recent in vivo or in vitro research that points on the BCG-based BC treatment limitations. While some limitations about the depth of the cancer cells in the urothelium were known, we found two relatively recent studies that provide more details on the manner. When comparing the underline results of the proposed model compared to the previous models of the same treatment, it is easy to see that the proposed model captures the limitation of the treatment while the other models are not. As such, formally, we introduce the statement:

Thus, overshooting the BCG threshold of a successful treatment protocol. Hence, the proposed model better agrees with the experimental results obtained by [69, 70] regarding the limitations of the BCG-based immunotherapy treatment of BC compared to previous models [26, 29, 32].

to the Discussion section, highlighting the clinical advantage of the proposed model.